# Arbuscular Mycorrhizal Fungi Are an Influential Factor in Improving the Phytoremediation of Arsenic, Cadmium, Lead, and Chromium

**DOI:** 10.3390/jof8020176

**Published:** 2022-02-12

**Authors:** Mohammad Reza Boorboori, Hai-Yang Zhang

**Affiliations:** College of Environment and Surveying and Mapping Engineering, Suzhou University, Suzhou 234000, China; m.boorboori@yahoo.com

**Keywords:** mycorrhizal fungi, metal(loid)s pollution, soil, plants

## Abstract

The increasing expansion of mines, factories, and agricultural lands has caused many changes and pollution in soils and water of several parts of the world. In recent years, metal(loid)s are one of the most dangerous environmental pollutants, which directly and indirectly enters the food cycle of humans and animals, resulting in irreparable damage to their health and even causing their death. One of the most important missions of ecologists and environmental scientists is to find suitable solutions to reduce metal(loid)s pollution and prevent their spread and penetration in soil and groundwater. In recent years, phytoremediation was considered a cheap and effective solution to reducing metal(loid)s pollution in soil and water. Additionally, the effect of soil microorganisms on increasing phytoremediation was given special attention; therefore, this study attempted to investigate the role of arbuscular mycorrhizal fungus in the phytoremediation system and in reducing contamination by some metal(loid)s in order to put a straightforward path in front of other researchers.

## 1. Introduction

### 1.1. Metal(loid)s Pollution and Phytoremediation

Soil is the basis of life and the most valuable ecosystem globally, which plays a crucial role in producing food and filtering air and water, and is a good platform for building our homes and cities [1,2]. Due to the rapid urbanization and industrialization of the world community, agricultural land is declining day by day while the demand for food is increasing [3]. Since chemical and physical disorders easily damage soils, thousands of years of human activity have left many contaminated soils worldwide [1,2]. One of the most critical soil pollutions is metal(loid)s pollution, which has become a global problem and poses a severe threat to human health and the environment [4]. Various activities such as mining, use of sewage sludge in agricultural lands, excessive use of pesticides and fertilizers containing metal(loid)s, smelting of metals, and irrational emission of waste in industrial activities increase the concentration of metal(loid)s in soil and surface water, and also affect concentrations underground [5,6]. Although metal(loid)s of lithogenic origin are found naturally, and in different concentrations in the soil [7], arsenic, chromium, lead, cadmium, copper, and zinc are among the most common metal(loid)s pollutants commonly found near industrial, mining, and agricultural sites [8,9].

The constant increase of metal(loid)s in the soil, and their non-degradability causes changes in biogeochemical cycles (including the impact on soil microorganisms and enzymatic activities) and ultimately causes imbalances in ecosystems [10,11]. Since metal(loid)s react chemically in different environments, this increases their mobility and bioavailability in the soil [12]; therefore, their uptake by plants is increased and thus causes the accumulation of metal(loid)s in plant seeds [13]. The accumulation of metal(loid)s in plants and contaminated drinking water causes them to enter the food chain of humans and animals and causes severe problems for their health [13,14] (Table 1). Therefore, there is an urgent need for the permanent removal of metal(loid)s or at least their long-term immobilization in soils and water in the present era [15].

Increasing soil contamination with metal(loid)s has prompted humans to think of ways to clear and restore these marginal, toxic, and contaminated soils to their reuse cycle [3]. Various techniques that are developed for this purpose include drilling contaminants, adding chemical regenerators, pumping contaminants, and physical stabilization by adding non-toxic materials [3,17]. Other methods used to remediate metal(loid)s contaminated soils include improving the soil physical and chemical traits by using mineral tailings, such as steel slag, zeolite, limestone, and fly ash [18]. It should be noted that steel slag is an alkali product that is a remnant of the steel industry and is composed of compounds of calcium, silicon, phosphorus, etc. [19], and due to its cheapness and availability, it can be used as a modifier of soils contaminated with metal(loid)s or fertilizers in agriculture [18,19].

Many methods of remediating soils contaminated with metal(loid)s are expensive and, at the same time, reduce the ability of soil to be reused in food production [20]. Recently, carbon-based materials to remediate contaminated soils have received more attention because it is more environmentally friendly and cheaper [21]. Biochar is one of these carbon-based materials produced by charring animal and plant biomass at a temperature of 300 to 600 °C under anaerobic conditions [22]. Biochar has alkaline properties, is stable, has a large surface area, high aromatic molecular structure, and can bind cations and prevent the migration and bioavailability of metal(loid)s in soil [23,24]. The addition of biochar to agricultural soils improves soil physical and chemical properties such as cation exchange capacity, soil pH, mineral retention, and adsorption capacity and positively affects plants growth [24]. Various studies showed that adding biochar to soils contaminated with metal(loid)s improves plant biomass and antioxidant enzymes and thus reduces the uptake of metal(loid)s by plants [25,26]. 

As mentioned, when soil ecosystems are contaminated with metal(loid)s, intervention is necessary to improve soil conditions. It is usually possible to stabilize metal(loid)s in the soil in the short term, but in the long run, it is costly [27]. Phytoremediation is one of the best ways to stabilize metal(loid)s in technosols in the long run [28], and it is the usage of plants to remove or move soil contaminants, in which plants are used to absorb or immobilize metal(loid)s in contaminated soils [29]. Phytoremediation is a cheap, effective, on site, and in general, eco-friendly process that does not require advanced engineering work and has raised great hopes for clearing soil ecosystems of metal(loid)s contaminants [30,31]. This technique can be conducted on vast scales and prevents metal(loid)s from penetrating groundwater aquifers [32,33]. 

According to this method, plants with high accumulation power in cooperation with soil microorganisms reduce the accumulation of metal(loid)s in soils [32]; on the other hand, this technique increases soil organic matter and microbial activity as well as reduces erosion processes and ultimately improves ecosystems [34]. The classification of phytoremediation involves the destruction, immobilization, inhibition, extraction of metal(loid)s, or a combination of these processes [35]. Most research on phytoremediation is related to plant extraction and stabilization. In the process of plant extraction, plants remove metal(loid)s from the soils by concentrating them in their shoots, while in plant stabilization, metal(loid)s become immobile in the roots of the plants [36].

This technique uses plants that are resistant to metal(loid)s that can accumulate high levels of metal(loid)s in their roots and shoots [37]. A plant’s tolerance to heavy metals depends on various factors such as the secretion of chelating agents in the rhizosphere, increased proline concentration, separation of metals, increased antioxidant enzymes activity, and the storage of metal(loid)s in the intracellular parts of the roots [38,39]. The resistance of plant species to metal(loid)s varies considerably between different plant families and genera, but in general vascular plants are slightly more tolerant than metal(loid)s [40]. Plants with high resistance to metal(loid)s are the best option for phytoremediation if they have an extensive root system and the ability to produce suitable aerial biomass (such as sunflower) [4,41]. On the other hand, nitrogen-fixing species that can survive in degraded environments are suitable examples of phytoremediation due to adding more organic matter and nutrients to the soil [42]. However, tree species have more potential to regenerate contaminated sites due to their ability to accumulate more metal(loid)s in plant tissues [43]. In aquatic ecosystems, wetland plants such as macrophytes are suitable tools for phytoremediation and reducing metal(loid)s pollution due to fast growth and high biomass production [44].

Plant stabilization is one of the essential methods in phytoremediation, which due to the accumulation of metal(loid)s in the roots of plants, prevents metal(loid)s spread and reduces their bioavailability in the soil [45]. Numerous studies showed that the cell walls of plants are the primary site of metal(loid)s immobilization [46]. Polysaccharides and proteins are the main components of the cell wall of plants, and the stabilization of metal(loid)s in the cell walls of plants can reduce metabolic damage of the cell [47]. The polysaccharide portion of the plant cell wall is composed of cellulose, hemicellulose, and pectin, and plants can adjust their polysaccharide content to cope with the stress of metal(loid)s [46,47]. Peroxidases are the major proteins in cell walls involved in plant responses to stresses and cell wall dynamics [48]. Additionally, the growth of plant roots adds secretions such as low molecular weight organic acids to the soil, which cause the weathering of soil minerals and the release of metals in the soil [49]. With the growth of plant roots, pores are created in the soil that provides pathways for rapid intercurrent movement and facilitates leaching of soil solution and preferential migration [50].

One of the positive points of phytoremediation is the restoration of the structure of the soil microbial community, which is a significant advantage over other soil remediation methods [51], and the success of phytoremediation is highly dependent on the presence of soil microorganisms [52]. Soil microorganisms improve plant growth and bioremediate contaminated soils by separating or degrading metal(loid)s [53]. Other factors that increase the efficiency of phytoremediation include the addition of cattle manure or earthworms and other microorganisms to contaminated soils [54,55,56].

### 1.2. Arbuscular Mycorrhizal Fungi (AMF)

AMF is one of the most common probiotic microorganisms, known as the most widespread fungi that coexist with plant roots [35,57]. About 240 different species of AMF belong to the Glomeromycota subfamily, which are found in almost all natural and agricultural ecosystems and coexist with most plants [58,59], to the extent that they are reported to coexist with 90% of terrestrial plants [60]. In addition, it was reported that various AMF, such as *Funneliformis mosseae*, *Rhizophagus irregularis*, and *Claroideoglomus claroideum* exist in wetland habitats [61], and various studies have shown that AMF can colonize the roots of plants such as rice [62,63].

Different studies showed that AMF biodiversity varies significantly in different environments, and their presence in different ecosystems results from different factors, including soil type, host plant, agricultural practices, and environmental conditions [64]. It should be noted that host plants also have a significant impact on AMF growth and efficiency [65]. AMF hyphae divide, grow, and form an extensive, dense network of hyphae in the soil [66]; in addition, AMF hyphae can drastically alter the structure and chemical properties of aggregates by releasing compounds such as proteins and polysaccharides [67]. In addition to creating stable aggregates resistant to wind and water erosion by binding soil particles together, AMF can also affect soil microbial activity and communities [68,69].

According to findings, AMF creates a direct relationship between plant roots and the substrate [70] and gives closer access to plant roots to absorb nutrients and water, among other soil microorganisms, thus significantly improving plant growth [71]. Research has also shown that the effects of AMF vary depending on the type of soil substrate and plant species [72]; however, AMF causes the establishment and survival of host plants in various environments such as saline soils, alkaline soils, agricultural lands, mine tailings, soils contaminated with metal(loid)s, etc. [73]. As mentioned, AMF improves plant nutrition and plant–water relationships by creating efficient coexistence with plants and increasing the uptake level and volume of available soil for the root system, thereby increasing plant tolerance to environmental stresses such as root pathogen damage [10,74,75]. 

AMF grows inside the roots of plants and allows them to get their necessary nutrients through networks of extra-radical mycelium (ERM) that spread in the soil [76]. AMF can dissolve, activate, and ultimately absorb nutrients such as phosphorus, nitrogen, and zinc [76,77], and in return, it receives carbohydrates and lipids from plants [9]. Researchers have shown that AMF has the most significant effect on plants’ phosphorus uptake [76], and it seems that up to 100% of the phosphorus required by plants is obtained through AMF phosphorus carriers [78]. AMF coexistence with plants causes phosphorus to be absorbed from a distance through fungal phosphate transporters such as GiPT [79,80] and effectively reaches plant roots through phosphate transporters (such as MtPT4) [81]. AMF also improves plant nutrition by increasing phosphate uptake from the soil because AMF increases the area of nutrient uptake through its hyphae [57], thus directly affecting plant growth [82], but on the other hand, increasing phosphorus uptake by the plant reduces this element in the soil solution [83]. Previous studies have shown that AMF symbiosis with plants has a positive effect on S uptake through regulating the expression of sulfate transporter genes (MtSULTR 1.1, MtSULTR2.1, MtSULTR 1.2, MtSULTR2.2, MtSULTR3.1, MtSULTR4.1, etc.) [81,84]; on the other hand, this coexistence directly affects S uptake through ERM activities [84].

As stated, AMF symbiosis with plants under stress conditions makes the plant more resistant to stress. Different studies have shown that exposed AMF spores to different stresses reduce ROS production with increasing antioxidant activities such as superoxide dismutases (SOD), glutathione (GSH), Vitamin B6, Vitamin C, and E [85,86]. AMF also activates plants antioxidant mechanisms such as glutathione peroxidase (GPx), ascorbate peroxidase (APX), catalyzes (CAT), and SOD [85]. In addition, AMF mycelium releases a particular glycoprotein called glomalin-related soil protein (GRSP), which helps improve soil conditions by forming complexes with heavy metals, helping to accumulate soil particles, increasing organic carbon content, and resulting in carbon sequestration [35,38,87]. Zhang et al. showed that AMF significantly increased the number of aggregates more prominent than 2 mm in contaminated soils [88]; a similar study also showed AMF-inoculation in Calcaric Regosol under drought stress, and proper irrigation increased the percentage of macro-particles larger than 5 mm [89].

Plants are the basis of phytoremediation, but various soil microorganisms such as AMF can significantly improve phytoremediation efficiency in different ecosystems [45,90], and this has shown that AMF naturally survives on high levels of metal(loid)s and helps plants withstand these contaminants [58]. The effect of AMF on soil improvement and phytoremediation depends on AMF species, metal(loid)s concentration, plant–metal tolerance, and metal(loid)s bioavailability [91]. AMF can also help the phytoremediation system by increasing immobilization, conversion, detoxification, and extraction of heavy metals [92]. Therefore, the use of plants that are more resistant to metal(loid)s and have more ability to accumulate more metal(loid)s, if they coexist with a suitable species of AMF (Glomeraceae family), have a significant effect on reducing pollution of heavy metal contaminated environments [85,93].

The growth of hyphae in environments contaminated with metal(loid)s is reduced [94], but extensive growth of AMF-mycelium by affecting the surface properties of aggregates increases the level of heavy metal uptake in phytoremediation systems, increasing the storage of metal(loid)s in soil and roots and thus reducing the transfer of metal(loid)s to aerial parts [95,96]. As mentioned, AMF has various solutions to reduce the bioavailability and biological absorption of metal(loid)s in the soil, including GRSP and antioxidant activities [86,97]. Phosphate, sulfhydryl, and other compounds in mycelium can prevent the transfer of metal(loid)s from the roots to the soil by converting their ions into forms such as oxalic acid extract, which have poor biological activity [98].

AMF-inoculation increases the resistance of host plants to metal(loid)s [99], which various factors can cause. The coexistence of host plants with AMF increases the absorption of water and nutrients such as nitrogen and phosphorus by plants and thus improves their growth [100]. Additionally, AMF, with the help of ERM, causes the deposition of polyphosphate complexes and the preservation of metal(loid)s in the roots of host plants, increasing the adaptation of plants to metal stresses [101]. The detoxification ability of AMF in plants largely depends on how AMF affects the host plant, the ability of AMF to precipitate metal(loid)s in the plant, the type of toxic metal(loid), and the availability of the toxic metal(loid) [25,98,99].

### 1.3. Arsenic (As)

Arsenic is one of the most toxic elements in nature, seriously endangering plants, animals, and even humans [102]; As is generally found in all crustal rocks but can be released due to natural factors or human activities in the environment [102,103]. Among the natural factors that cause the release of As in nature are volcanic activity and weathering of rocks [104], but the human factors that cause the release of As in the environment are mining, fossil fuels, tannery, pesticides, herbicides, and chemical fertilizers [102,103,104]. The most important cause of arsenic poisoning is contaminated groundwater found in Bangladesh, Italy, Argentina, Hungary, India, China, Mexico, Chile, and the United States [105,106].

Gupta et al. reported that As affects growth and productivity due to the morphological, biochemical, and physiological changes it causes in plants [104]. As in plants reduces transpiration rate and leaf water potential, chlorophyll content (Chl), nutrient uptake, CO_2_ stabilization rate, photosystem II activity, photosynthesis rate, heat loss capacity, carbon splitting, and sugar metabolism [107,108,109,110,111,112]; also, one of the most dangerous biochemical effects of As is the production of intracellular reactive oxygen species (ROS), which causes irreversible damage to DNA, lipids, carbohydrates, and proteins [107]. Symptoms of As in plants include reduced germination, biomass, leaf area, number of leaves, and yield [113]. The reaction of different plant species to As is different; for example, the yield of potatoes (*Solanum tuberosum* L.) decreases at a concentration of 300 mg/kg As in soil [114], while in the case of soybeans (*Glycine max* L.), it is 35 mg/kg As in soil [115]. However, this concentration for rice (*Oryza sativa* L.) is equal to 25 mg/kg As in soil [116], which can significantly reduce rice yield from 7–9 tons to 2–3 tons per hectare [117] (Table 1). Arsenic is a class 1 human carcinogen [118] that is not only transmitted to the human body through food and drinking water but also its prolonged inhalation causes poisoning in humans [105], which can cause skin ulcers, cardiovascular problems, lung and bladder diseases, cancer, and eventually death [105,119]. Therefore, finding effective ways to remove more arsenic from contaminated environments is essential.

Arsenic exists in inorganic and organic forms in nature, and its organic species are more toxic and mobile [120]. The most common types of As that are uptaken by plants are arsenate, arsenite, MMA, and DMA [113] (Table 2). Although different forms of arsenic are present in the environment simultaneously, plants receive it from the soil with a special preferential system (arsenite, arsenate, dimethylarsinate, and methylarsonate) [113]. It should be noted that As(V) is predominant in oxidizing media, and As(III) is more prevalent in reduced environments [120]. The most common way As enters plants is through roots, but the distribution of As in plant organs is very different so that the highest amount of As accumulates in plant roots and then in leaves, shoots, pods, and seeds [121]. Various studies showed that As(V) enters plants through phosphate transporters (Pht1 and Pht4) and As(III) through silicon transporters (OsLsi1 and OsLsi2) [122,123]. Arsenic enters plant cells mainly in the form of As(III) and As(V), but eventually As(V) is also catalyzed and converted to As(III) by arsenate reductase enzyme and is pumped through special cells transmitters or stored in vacuoles [124,125].

Since AMF is naturally present in As-contaminated environments [127], it can be an important component of increasing the efficiency of phytoremediation methods [128]. Research has also shown that arsenic can damage AMF and inhibit the primary stages of its development cycle [129], which can ultimately reduce mycorrhizal colonization [85]. Various studies showed that the prevalence of different species of AMF varies in arsenic stress environments [130], for example, Gonzalez-Chavez et al. found that *Glomeraceae* and *Acaulosporaceae* are predominant species in contaminated environments in Brazil [131]; however, *Glomeraceae* and *Glomus* are the predominant family and genus in arsenic-contaminated ecosystems [132].

Spagnoletti et al. reported that mycorrhiza plants are more tolerant of As toxicity, which may be due to various factors [85], which we will address below. AMF-inoculation increases plant biomass and dilute As in plants, and as a result, it increases the plant’s resistance to As toxicity [85,100]. Studies showed that AMF-inoculation increases the uptake of nutrients, including phosphorus (P), nitrogen (N), and magnesium (Mg), by plants [104,133]. It can also result in increased photosynthetic pigment concentration, higher Hill reaction activity, an optimum chlorophyll a/b ratio, and higher photosystem II activity [104]. Increasing the absorption of CO_2_ and improving the metabolism of essential carbohydrates are other positive roles of AMF in plants [104]. All of the above factors increase the resistance of mycorrhiza plants in As-contaminated environments.

On the other hand, the effect of AMF on the uptake, displacement, and speciation of As is well identified [134]. Research showed that As is absorbed by AMF hyphae (through RiPT and GiPT gene) [135,136], AMF reduces As(V) to As(III) (through RiarsC gene) [136], and finally, As is released through RiArsB, ATPase pump, and GiArsA into the soil [135,136]. It was shown that AMF can evaporate and methylate inorganic As through RiMT-11 into a wide range of organic As [134,137]; also, AMF causes more DMA release, especially when high concentrations of As(V) are present in the environment [134]. Numerous reports showed that coexistence with AMF has increased As evaporation and methylation as well as increased the As(III) to As(V) ratio in various crops, including rice and alfalfa [134,138,139].

Another AMF tool to counter As toxicity is GRSP secretion in the soil [140], and given that GRSP has a high amount of iron, it can produce AsIII-FeIII and ultimately immobilize As in the soil through bio-adsorption [85,140]. Spagnoletti et al. showed that with increasing As concentration in soil, GRSP content in the soil also increases [141] (Figure 1). Other benefits of AMF include helping to increase the activity of plant antioxidants in As-contaminated environments [85]. In this case, Spagnoletti et al. reported an increase in SOD, CAT, GPX, and GSH activities in AMF-inoculated soybeans in arsenic-contaminated soils [85,135]. In addition, the researchers found that AMF regulated the uptake of As and some other elements into plant roots by affecting the protein synthesis of channels related to As uptake and other elements such as P [105,135,142]. Christophersen et al. found that AMF reduced the expression of MtPht1 and MtPht2 genes, which are involved in the uptake of heavy metals into the root membrane of *Medicago truncatula*, but instead improved the expression of the MtPT4 gene, which carries P [142]; Li et al. Additionally, reported similar results for alfalfa [139].

So far, only a handful of arsenic hyperaccumulator plants are identified, some of which are listed below: Pteris vittata, Pteris ryukyensis, Peris biaurita, Pteris capadogramatica, Pteris longifolia, Pteris fauriei, Pteris umbrosa, Pteris capadograma, Pteris quadriaurita, Pteris cretica, Pteris oshimensis, and Pteris aspericanlis [132]. Therefore, recognizing more plant species that are highly compatible with As and also suitable AMF species with which plants coexist more significantly impacts the phytoremediation of As-contaminated environments [45].

### 1.4. Cadmium (Cd)

Cd is an unnecessary element for animals and plants and is naturally present in low concentrations in soil and rocks [143,144]. This element is stable and does not decompose, so it is one of the most common contaminants in agricultural lands, especially in China. If the concentration of Cd in the soil exceeds 0.5 mg kg^−1^, it will cause damage to plants and animals [144,145,146] (Table 1). In nature, the increase in Cd pollution is mainly due to human activities, including mining, metal smelting processes, Cd-rich phosphate fertilizers, industrial effluents, and fuel production [147,148]; the International Agency for Research on Cancer categorized it as a group 1 carcinogen [149]. Due to the high solubility of Cd, heavy rains, field irrigation, fine soil particles, and preferential flow cause the leaching of Cd from the surface layers of the soil to the subsoil, which is a factor in increasing Cd pollution in groundwater [150,151,152].

Cd is transported to plant tissues due to its common pathway with essential elements such as potassium (K) and calcium (Ca) [153]. High accumulation of it in plant tissues is toxic and reduces water and nutrient uptake, reducing plant growth, chlorosis, and eventually, the plant dies [154]. Plants have several important strategies for increasing their tolerance to Cd contamination, which can generally be divided into four categories: reduction of cell membrane transmission, Cd attachment to the cell wall, chelation, and compartmentalization [82]. Therefore, finding solutions to improve plant performance and increase their resistance to cadmium can be a good way to absorb, stabilize, accumulate, and reduce the toxicity of this element in contaminated environments and thus prevent its penetration into groundwater or the food chain [99,155]. One of these strategies is the coexistence of plant roots with soil microorganisms, especially AMF, which reduces Cd’s toxicity, bioavailability, and environmental migration in the soil [49]. 

Although some studies have suggested that high concentrations of Cd inhibit mycorrhizal colonization, an important part of the research showed that different Cd concentrations do not significantly affect AMF colonization [4,156]. Additionally, different researches have shown that different species of AMF have different effects on the uptake of Cd by plants or its stabilization in soil [4,157,158]. For example, Hassan et al. showed that *Rhizophagus irregularis* in contaminated environments compared with *Funneliformis mosseae* caused more uptake of Cd by the sunflower and, as a result, increased its accumulation in the shoots of this plant. In contrast, *Funneliformis mosseae* inoculation increased the deposition of Cd in the soil [4]. Jiang et al. also reported that *Glomus versiforme* significantly reduced Cd accumulation in shoots and roots, while *Rhizophagus intraradices* increased Cd concentrations in roots and decreased cadmium concentrations in shoots [159].

AMF increases the resistance of host plants to Cd toxicity by improving photosynthesis, antioxidant enzymes, water and nutrition absorption, and growth [88]. Various studies showed that AMF inoculation increased the activity of SOD, CAT, APX, peroxidases (POD), and the total soluble protein content of plants tissues grown in Cd-contaminated environments while decreasing malondialdehyde (MDA) content in plants under similar conditions [160,161]. In addition, AMF helps increase the growth and biomass of the host plant by helping to increase the root length of plants and improve photosynthesis conditions, resulting in greater resistance of plants in Cd-contaminated environments [49,162]. He et al. also showed that AMF hyphae are prevented from N and P leaching in the soil, making them more accessible to plants [49] (Figure 2). Better absorption of nutrients and improved water absorption are other factors to increase the resistance of plants to Cd [160], which were already mentioned.

In addition, AMF hyphae are easily intertwined on the soil particle surfaces and reduce their bioavailability and migration by adsorption and stabilizing Cd ions [96,163]. In addition, GRSP, which is the result of the degradation of AMF mycelium, can form a complex with Cd and significantly increase its uptake by soil particles [97,164]. On the other hand, AMF increases the formation of coarse-grained soil, which, with other factors mentioned above, reduced the Cd in porous water, thus reducing Cd leaching to the depth of soil and groundwater [49,105] (Figure 2).
Figure 2The role of AMF in cadmium contamination in the plant, water, and soil. Nomenclature is as proposed by Gunathilakae et al. [165].
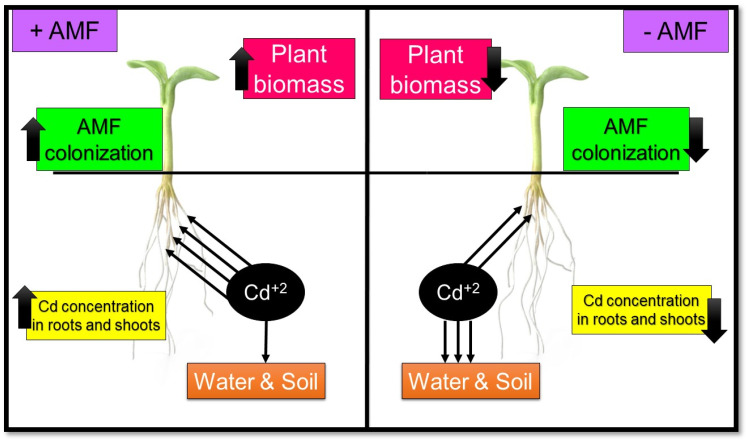



The researchers showed that inoculation of some species of AMF in different plants increases the uptake of cadmium by the roots of plants, in which case, depending on the AMF species, increases the accumulation of cadmium in the roots of plants or its transfer to the shoot [4,49], which in any case causes reduced Cd contamination in soil and improves phytoremediation in contaminated environments [166] (Figure 2). For example, Audet and Charest showed that *Rhizophagus irregularis* inoculation allows the sunflower plant to be used as a Cd accumulator [166].

Numerous studies showed that inoculation of AMF with the use of biochar and steel slag can firstly increase the resistance of plants to cadmium contamination and secondly increase the rate of stabilization and accumulation of cadmium in the soil and roots of plants [5,25]. In a study on the corn plant, Hu et al. found that inoculation of three different species of AMF, including *Glomus versiforme*, *Funneliformis mosseae,* and *Rhizophagus intraradices* with the application of steel slag in Cd-contaminated environments not only increased plant growth and decreased Cd uptake into plant organs, but also caused increased pH and total glomalin content in the soil [5]. Liu et al. also observed that AMF-inoculation and biochar application in maize simultaneously increased plant growth, increased antioxidant activity such as SOD, CAT and POD, and decreased Cd concentration in different plant organs. This combination treatment also caused increased soil pH and cadmium stabilization and finally decreased cadmium bioavailability in soil [25].

### 1.5. Lead (Pb)

Pb is one of the most toxic heavy metals in nature, and due to its high stability and lack of decomposition, it pollutes nature and accumulates in plants and living organisms [167]. This heavy metal is one of the most common pollutants of the present age, which is very dangerous due to the soil and climate pollution. The removal of this pollution helps maintain the health of humans, other creatures, and nature [145,168]. Pb is naturally distributed in the Earth’s crust, and due to human activities, has become the most widespread toxic metal in nature [169]. Among the human factors that cause environmental pollution to Pb are mining, smelting metals, and industrial effluents [147]. Other factors that lead to the spread of Pb pollution in the environment include the performance of small companies in some countries, including Pakistan, which repair obsolete lead-acid batteries and, because they are not able to comply with strict environmental regulations, discharge wastewater directly into the soil, waterways, and the surrounding environment, which causes severe environmental pollution with Pb. This danger is exacerbated when Pb-contaminated water is used to irrigate fields [138].

Increasing the concentration of Pb in the environment causes adverse effects such as the inhibition of seed germination, slow plant growth and stunting, reduced metabolism and photosynthesis, accumulation of Pb in plant tissues, chlorosis, and eventually plant death [170,171]. Therefore, the increase of Pb in the environment is a serious threat to food security [172]. On the other hand, prolonged exposure of children to Pb contamination adversely affects the development of their nervous system and is likely to lead to mental retardation [173]. Other side effects of Pb in humans include anaemia and kidney disease, to the extent that research showed that human kidney cells are destroyed by high concentrations of Pb [174,175]. Therefore, phytoremediation can reduce Pb’s adverse effects on human, animal, and plant communities.

Salazar et al. showed that AMF has suitable Pb accumulation mechanisms in their spores and mycelium, but the amount of Pb uptake may depend on AMF species and host plants, and in general, Glomeraceae is the most important diverse species of AMF present in Pb-contaminated soils [176]. AMF, on the other hand, helps prevent the transfer of Pb to the shoot by helping to increase the accumulation of Pb in plant roots [70]. Another important role of AMF excretes total glomalin-associated soil protein (TGSP), which increases the retention of Pb in soil, reduces the bioavailability of it, and thus reduces the toxicity of Pb to plants [35,98,177].

The researchers also showed that AMF increases plant resistance in lead-contaminated environments by increasing plant nutritional efficiency and improving the antioxidant defence system [35,98]. Chen et al. showed that *Populus euphratica* inoculation with *F. mosseae* in Pb-contaminated environments significantly increased SOD and CAT activities [178]; however, Spagnoletti et al. also reported that *Cichorium intybus* inoculation by *R. irregularis* increased SOD and CAT activity in Pb infected environments [85]. Research also showed that the inoculation of plants with AMF by increasing the content of polysaccharides in the hemicellulose and pectin of the cell wall, and increasing peroxidase activities in the cell walls and thus increasing Pb fixation in the root cell wall of host plants, makes them more resistant to Pb toxicity [105]. Zhang et al. showed that the expression of MtPrx05 and MtPrx10 genes related to cell wall polysaccharide cross-linking was increased by AMF-inoculation in Pb-contaminated media [105] (Figure 3).

In recent years, scientists have noted the positive effect of inoculating plants with AMF and adding earthworms, biochar, cow manure, lignin, and steel slag to increase uptake and stabilize Pb from contaminated environments [5,24,179,180]. They found that combining AMF with other factors improves phytoremediation in Pb-contaminated environments through increasing the uptake of nitrogen, phosphorus, potassium, and iron, reducing Pb transfer from root to shoot, increasing root colonization, improving soil pH, further plant growth, and increasing soil glomalin content [5,24,179,180].

### 1.6. Chromium (Cr)

In addition to being one of the most abundant elements in the Earth’s crust, Cr is also one of the most dangerous heavy metals [181]. The abundance of Cr in the soil indicates an environmental problem that is most likely originated from human activities, including chemicals used in agriculture, paint and leather industries, alloy production, and stainless steel [182,183]. It can be said that India’s tanning industry alone imports between 2000 and 32,000 tons of Cr into the environment annually [88]. Cr is very dangerous because it does not decompose chemically or biologically, and its concentration in living organisms increases as it moves along the food cycle, turning it into a dangerous environmental contaminant in soil, water, and air [182]. 

Cr usually exists in two stable forms, trivalent [Cr(III)] and hexavalent [Cr(VI)], both of which can be exchanged through the precipitation/dissolution, oxidation/reduction, and adsorption/desorption processes [184]. Cr(III) is the most abundant and stable form of Cr [181], which is non-toxic and is usually immobile and insoluble in water [185]. Cr(III) is susceptible to adsorption on the soil surface or deposition in chromium hydroxide form in slightly acidic or alkaline environments [60], which plants can not easily absorb [183]. Cr(VI), on the other hand, is a class A carcinogen substance that can kill living cells [183]. This substance, which is completely soluble in water in the pH range, is highly mobile and is usually present in neutral to alkaline soils, mainly in the form of a chromate anion (CrO^2−^_4_) or relatively sparing chromate salts such as PbCrO_4_, BaCrO_4_, and CaCrO_4_ [60,183]. It should be noted that Cr(III) is considered an essential human substance that can interfere with cholesterol, glucose metabolism, and increased insulin secretion [181], but Cr(VI) is highly toxic by inhalation and, in high concentrations, cause adverse effects such as renal failure, hemolysis and liver failure [183,185]; therefore, the permissible dose in water is 8 micrograms per litre for Cr(III) and 1 microgram per litre for Cr(VI) [186]. 

Cr is the most toxic pollutant that negatively affects plants’ performance and metabolic activity [187]. Cr(VI), as a strong oxidizing with redox potential between 1.33 to 1.38, causes rapid production of reactive oxygen species (ROS) such as superoxide and hydroxyl radicals [188], and its negative effects on plants include changes in membrane structure and root damage, carbon uptake, antioxidant defence activity, nutrient uptake, DNA damage, ion transport imbalance, reduced photosynthesis and growth, and eventually plant death [39,186,189,190]. Plants have different mechanisms for combating the toxicity of Cr, and the most important of which is the chemical reduction of Cr(VI) to Cr(III), which can be carried out enzymatically and non-enzymatically [191]. The addition of salts containing Fe(III), animal manure, or organic acids to the culture medium helps the plants in this direction [9]. Therefore, plants that accumulate Cr, such as *Prosopis laevigata*, *Spartina argentinensis,* and *Amaranthus dubius*, can convert Cr(VI) to Cr(III) and prevent chromium erosion and leaching in the soil [32,92]. Due to the low cost, phytoremediation can be a good strategy for improving and cleaning Cr-contaminated environments [9].

According to studies, factors such as AMF play an important role in regulating Cr uptake and detoxification of plants [192], but the effect of this factor depends on the type of plant, the type of fungus and soil conditions [182]. Research showed that AMF absorbs more chromium through its various structures (such as hyphae, ERM, and spores), and complexing Cr with histidine or phosphate prevents them from being transferred to plants [101,182]. AMF can also use various strategies such as helping to absorb nutrients (P and N) to increase plant resistance to Cr stress and prevent severe damage or death due to Cr toxicity [182]. Another important role of AMF is to stabilize Cr in plant roots through external and internal radical mycelium and ultimately help reduce its displacement to plant stems [101]. In addition, BGlomalin-secretion by AMF can immobilize Cr [164]. Due to their similar chemical structure to Cr(VI) and P, these potentially compete during the adsorption process by plants, and since AMF increases P uptake by plant roots, they can play an important role in reducing the uptake of Cr(VI) [193]. Gil-Cardoza et al. also found that AMF could help detoxify Cr(VI) by reducing it to Cr(III) through ERM [193].

As mentioned, plant coexistence with AMF may help detoxify Cr by improving plant mineral nutrition and producing important metabolites such as sulfur compounds [81]. Since sulfur metabolites can combine with Cr through thiol groups to reduce the toxicity of Cr in microorganisms and plants [194], increasing the uptake of S by plants can provide the conditions for reducing the toxicity of Cr [195]. Among the S metabolites are glutathione (GSH), phytochelatins (PCs), and cysteine (Cys), which can act as non-enzymatic antioxidants in the elimination of ROS induced Cr [81,196]. Wu et al. also showed that AMF inoculation increases the expression of sulfate transporter genes with high affinity (MtSULTR1.1 and MtSULTR1.2) in plant roots and thus increases S uptake, which ultimately increases Cr(VI) detoxification [81,101].

## 2. Conclusions

As observed, AMF-inoculation in various forms increases phytoremediation efficiency in environments contaminated with arsenic, cadmium, lead, and chromium. AMF increases the accumulation of these metal(loid)s in the soil and roots of plants, prevents them from washing deeper into the soil and penetration into groundwater, and increases the resistance of plants to the high toxicity of these metal(loid)s. Increasing awareness of ways to improve the performance of AMF in phytoremediation, especially in the case of lead, about which there is limited information, can introduce phytoremediation as one of the most practical and cheapest ways to improve contaminated sites in many parts of the world. It should be noted that knowing more about plants that accumulate metal(loid)s and AMF species that coexist better with plants will help in this way.

## Figures and Tables

**Figure 1 jof-08-00176-f001:**
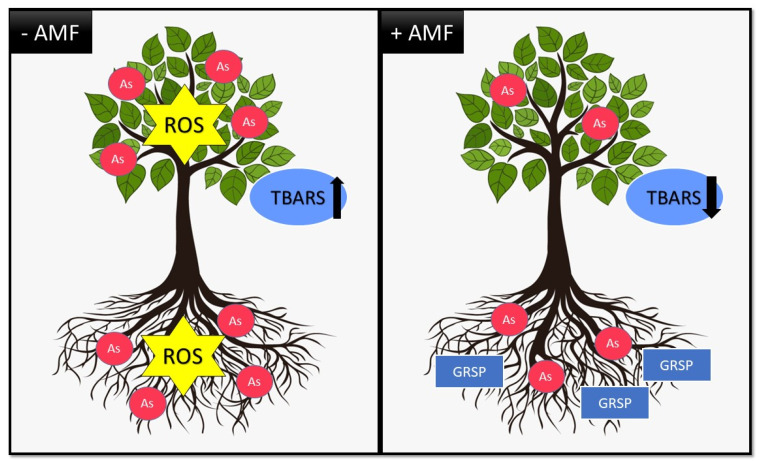
An overview of the role of AMF in increasing plant tolerance to arsenic contamination. Nomenclature is as proposed by Spagnoletti et al. [85]. AMF: arbuscular mycorrhizal fungi; As: arsenic; ROS: reactive oxygen species; GRSP: glomalin-related soil protein; TBARS: thiobarbituric acid-reactive species.

**Figure 3 jof-08-00176-f003:**
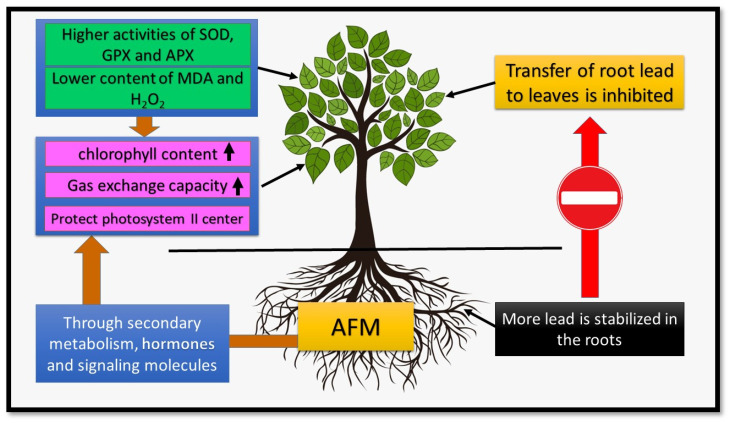
AMF inoculation role in plant tolerance to Pb. Nomenclature is as proposed by Yang et al. [91].

**Table 1 jof-08-00176-t001:** Range of some metal(loid)s in terrestrial plants and regulatory standards for them in food and drinking water in different countries.

Metal(loid)s	Content Measured in Different Plants (µg/g DW)	WHO * (mg/kg)	Canada (in Row Herbal Materials) (mg/kg)	China (Herbal Material) (mg/kg)	India	
					Food (mg/kg)	Water (mg/L)
Arsenic	0.02–7	Nil	5	2	1.1	0.05
Cadmium	0.1–2.4	0.3	0.3	1	1.5	0.01
Lead	1–13	10	10	10	2.5	0.1
Chromium	0.2–1	Nil	2	Nil	20	0.05

Nomenclature is as proposed by Gjorgieva Ackova D [16]. * World Health Organization.

**Table 2 jof-08-00176-t002:** The list of various As species in nature.

Arsenic Compounds	Acronyms	Chemical Formula
Arsenate	As (V)	As(O^−^)_3_
Arsenite	As (III)	O=As(O^−^)_3_
Methylarsonate	MMA	CH_3_AsO(O^−^)_2_
Dimethylarsinate	DMA	(CH_3_)_2_AsO(O^−^)
Trimethylarsin oxide	TMAO	(CH_3_)_3_AsO
Tetramethylarsonium ion	TETRA	(CH_3_)_4_As^+^
Arsenobetain	AB	(CH_3_)_3_As^+^CH_2_COO^−^
Trimethylarsoniopropionate	TMAP	(CH_3_)_3_As^+^CH_2_CH_2_COO^−^
Arsenocholine	AC	(CH_3_)_3_As^+^CH_2_CH_2_O^−^
Dimethylarsinoylacetate	DMAA	(CH_3_)_2_(O)As^+^CH_2_COO^−^
Dimethylarsinoylpropionate	DMAP	(CH_3_)_2_(O)As^+^CH_2_CH_2_COO^−^

Nomenclature is as proposed by Boorboori et al. [126].

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
