# Peer review of "Arbuscular Mycorrhizal Fungi Are an Influential Factor in Improving the Phytoremediation of Arsenic, Cadmium, Lead, and Chromium"

_jof, 2022, doi:10.3390/jof8020176_

Round 1

Reviewer 1 Report

The manuscript titled "Arbuscular mycorrhizal fungi is an influential factor in improving the phytoremediation of arsenic, cadmium, lead, and chromium" is an excellent piece of work and has lot or relevence in todays problems of soil pollution. This manuacripr cam be considered for piblication after necessary minor revision.

  1. Author need to add few more hevay metals present in the soil and their phytoremediation by AMF.
  2. One paragraph need to be added in the introduction, about the heavy metal consequences on agriculture.
  3. Please eloborate is there any synergism of heavy metal toxicity, when one than one metal is present by consulting the literature.
  4. Author need to check once again the chemical formulla and units.
  5. Few grammitical   mistakes are there, and author need have a thorough look on whole manusript.

Author Response

Dear Sir/ Madam

Thank you so much for your helpful comments

we have modified the manuscript according to your comments, you can find them below:

  • This manuscript just focused on the phytoremediation of arsenic, cadmium, lead, and chromium, but thank you very much because of your suggestion, surly we will write about the effect of AMF on other heavy metals .
  • we already published about the antagonism between some chemicals and heavy metals and the effect of hevy metal in agriculture which some of them were published in the MDPI journal

https://doi.org/10.3390/app11178090

https://doi.org/10.3390/plants10102210

https://doi.org/10.3390/agronomy11081532

  • All the chemical formulas and units were re-checked.
  • All the manuscript was re-checked, and some grammar mistakes were modified.

Thanks again for all your helpful comments, and if there are any more comments, I will be happy to know about them.

Best regards

M. Boorboori

Reviewer 2 Report

General comment

This manuscript compiles a big amount of information, which may be of interest for someone that wants to start working on this subject. I will be happy to revise it again after some changes are made. In my opinion, the manuscript can be improved by:

  • Checking English and phrase structuring thoroughly
  • Redesigning the way sections are presented (only introduction and conclusion is too limiting)
  • Working on figure captions

Specific comments

Title

Shouldn’t it be “Arbuscular mycorrhizal fungi are” instead of “Arbuscular mycorrhizal fungi is…”

Line 10

One of the most dangerous contaminants of the present age is metal(loid)s contamination

Please revise

finding suitable solutions to reduce metal(loid)s further

To reduce their concentration…? Clarify

Line 18

Arbuscular -> arbuscular

Line 19

“and reduce contamination of some metal(loid)s” -> “and in reducing contamination by some metal(loid)s”

Line 37

Delete the “but”, since the section starts with “although”

Line 44

“be increased” -> “is increased”

Line 45

“Accumulation of metal(loid)s in plants, and contaminated drinking water causes them to enter the food chain of humans and animals and cause severe problems for their health” -> Accumulation of metal(loid)s in plants and contaminated drinking water causes them to enter the food chain of humans and animals and causes severe problems for their health

Table 1

“metal(loid)s” -> Metal(loid)s

WHO needs to be written in full in the caption

What are the Indian standards? Median values? Limits defined by law?

What data was used to build the table? Include the information in the caption

Line 62

Revise

Line 68

In production of what? Food?

Line 86

“friendly method, which no need for advanced engineering”

Revise

Line 92

“however, on the oher hand, this technique is increased soil organic matter…”

Revise

Line 93

Revise this phrase. It is a bit confusing.

Line 106

“Plants with high resistance to metal(loid)s if with an extensive root system…”

Revise

Line 141

“Probiotics” -> “probiotics”

Line 177

“Phosphate”

Line 188

SOD, GSH and other enzymes need to have their name written in full along with the abbreviations

Line 191

Is glutamate a glycoprotein? Isn’t it an amino acid? Also, I did not find in reference 38 any mention towards this compound

Line 212

“…and roots; Thus, reducing” -> and roots, thus reducing

Line 229

“endangered” - endangering

Figure 1

Abbreviations should be explained in the caption.

Line 385

MDA  needs to be also written in full

Lines 441, 444

“Increased” -> increased

Line 484

“However” -> however

Line 562

Such -> such

Line 591

Revise

  1. Patents

Delete

Author Response

Dear Sir/ Madam

Thank you so much for your helpful comments

we have modified the manuscript according to your comments, you can find them below:

  • The title was modified.
  • The sentence in line 10 was rewritten.
  • Line 13-15 was rewritten.
  • In line 18, arbuscular was added.
  • Line 19 was modified.
  • Line 37, “But” was deleted.
  • In line 44, “is increased” was added.
  • Line 45-47 were modified.
  • In Table 1, “Metal(loid)s” was added
  • “World Health Organization” was added to the caption of Table 1.
  • In Table 1, I removed “standards” in front of “India”, and made it the same as other countries. In India, standards for metalloids are for food and water, but in other groups, it is for plants.
  • The source of data for Table 1 was added in the caption of the table.
  • Line 60-62 was rewritten.
  • Line 68, “Food,” was Added.
  • Line 86 was modified.
  • Line 92 and 93 rewrote.
  • Line 106-108 rewrote.
  • Line 141 “probiotics” modified.
  • Line 177 “Phosphate” modified.
  • Line 188, the full name of SOD and GSH added.
  • Line 191 “GRSP” is “glomalin-related soil protein,” and it was added and it is a glycoprotein
  • Line 212, “and roots, thus reducing” replaced.
  • Line 229, “endangering”
  • Abbreviations added to Figure 1.
  • In Line 385 MDA full name added.
  • “increased” modified in line 441 and 444.
  • “however” replaced in line 489.
  • “such” replaced in line 567.
  • Line 596 rewrote “especially in the case of lead, about which there is limited information”.
  • Because I added the name of one of my colleagues as an author, so i kept

Thanks again for all your helpful comments, and if there are any more comments, I will be happy to know about them.

Best regards

M. Boorboori